# CONTINUAL LEARNING BY REUSE, NEW, ADAPT AND SKIP: A HIERARCHICAL EXPLORATION-EXPLOITATION APPROACH

## ABSTRACT

To effectively manage the complexities of real-world dynamic environments, continual learning must incrementally acquire, update, and accumulate knowledge from a stream of tasks—without suffering from catastrophic forgetting of prior knowledge. While this capability is innate to human cognition, it remains a significant challenge for modern deep learning systems. At the heart of this challenge lies *the stability-plasticity dilemma*: the need to balance leveraging prior knowledge, integrating novel information, and allocating model capacity adaptively based on task complexity. In this paper, we propose a novel exemplar-free class-incremental continual learning (ExfCCL) framework that addresses these issues through a Hierarchical Exploration-Exploitation (HEE) approach. Our method centers on two key subsystems: (i) a HEE-guided neural architecture search (HEE-NAS) that enables a learning-to-adapt backbone via four primitive operations—reuse, new, adapt, and skip—thereby serving as an internal memory that dynamically updates selected components across tasks; and (ii) a task ID inference mechanism, which utilizes an external memory of task centroids to select the appropriate task-specific backbone and classifier during testing. We term our overall framework **CHEEM** (Continual Hierarchical-Exploration-Exploitation Memory). CHEEM is evaluated on the challenging MTIL and Visual Domain Decathlon (VDD) benchmarks using both Tiny and Base Vision Transformers. It significantly outperforms state-of-the-art prompting-based continual learning methods, closely approaching full fine-tuning upper bounds. Furthermore, it learns adaptive model structures tailored to individual tasks in a semantically meaningful way, demonstrating its effectiveness in exemplar-free continual learning scenarios.

## 1    INTRODUCTION

Developing continual learning machines is a key objective in Artificial Intelligence (AI), aiming to replicate human-like adaptability and the ability to learn-to-learn, enabling proficiency in streaming tasks. Despite their advances, state-of-the-art Deep Neural Networks (DNNs) still lack true biological intelligence in the realm of continual learning from streaming tasks in dynamic environments, which requires the continual acquisition, update, and accumulation of knowledge while mitigating catastrophic forgetting of previous tasks (McCloskey & Cohen, 1989; Thrun & Mitchell, 1995), referring to the stability-plasticity trade-off.

Recently, continual learning using Vision Transformers (ViTs) (Dosovitskiy et al., 2021) has witnessed promising progress, primarily explored through the lens of prompt-tuning or prefix-tuning (Wang et al., 2022d;c;a; Smith et al., 2023). Of particular interest is Exemplar-free Class-incremental Continual Learning (ExfCCL), where the raw data (or latent features of samples) of old tasks are not available in learning a new task, and task IDs of testing samples are unknown at inference.

Denote by $\mathcal{T} = \{1, 2, \cdots, t, \cdots, N\}$ a stream of $N$ tasks in continual learning, where each task $t$ consists of a training set $D_t^{train}$ and a testing set $D_t^{test}$. Task 1 is assumed to train a ViT sufficiently well (Wang et al., 2022d;c;a; Smith et al., 2023), consisting of a backbone and a head classifier, $(F_1, H_1)$. In this paper, we make no restrictive assumptions in the remaining tasks, $\mathcal{T} \setminus \{1\}$, regarding the task nature, order, or number of classes in streaming tasks (either per task or in total). For example, there are 11 diverse, streaming tasks in the MTIL benchmark (Fig. 2a). Let task $t$ have $C_t$ classes, so

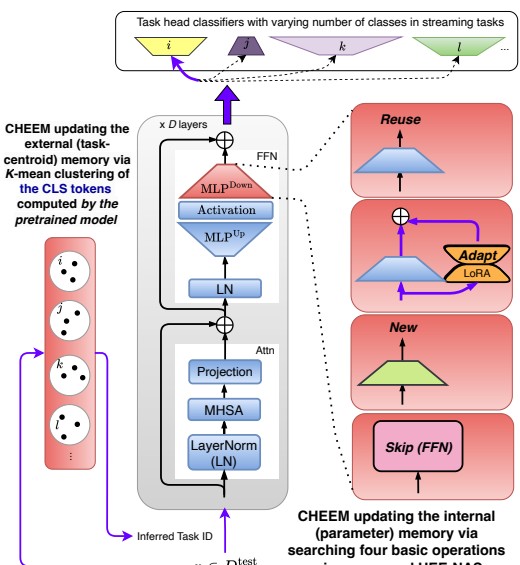

Figure 1: Illustration of the proposed CHEEM. A pretrained and frozen ViT model such as ViT-Base (Dosovitskiy et al., 2021) or DEiT-Tiny (Touvron et al., 2021) is structurally and dynamically updated to learn internal (parameter) memory for streaming tasks in continual learning, and is also used in maintaining the external task-centroid memory of CHEEM. CHEEM learns the internal parameter memory via hierarchical exploration-exploitation sampling based NAS using four operations (Reuse, Adapt, New and Skip) for a selected component such as the MLP$^{\text{Down}}$ layer. We also test placing CHEEM at the projection layer in the 'Attn' block.

by time $T$ we have observed tasks $1, \ldots, T$ with a total of $\sum_{t=1}^{T} C_t$ classes. Denote by $(\mathcal{F}_t, \mathcal{H}_t)$ the continually learned ViT backbones and head classifiers after task $t$ (for $t \geq 1$) in ExfCCL. We have two design choices as follows:

**Static Backbone vs Dynamic Backbone:** For a static backbone, $\mathcal{F}_t \equiv F_1$ ($\forall t \geq 1$), and the continual learning capability is achieved through prompt-tuning or prefix-tuning (Wang et al., 2022d;c;a; Smith et al., 2023), which often entails large pretrained ViTs to accommodate all streaming tasks of diverse nature and thus pays the computational cost "blindly" across tasks totally ignoring their difficulty levels.

For dynamic backbones, $\mathcal{F}_t$ is the super backbone structurally and dynamically updated from the pretrained model $\mathcal{F}_1 = F_1$, and $F_1 \subset \mathcal{F}_t$ ($t > 1$). Let $\Theta_t = \mathcal{F}_t \setminus \mathcal{F}_{t-1}$ be task-specific backbone parameters to the task $t$, which we term **the internal parameter memory** in ExfCCL to exploit task synergies.

**Task-Agnostic Head vs Task-Specific Head at Inference:** For the task-agnostic head at inference, there often exists a discrepancy between training and inference. During training, for a task $t$, only the segment head $H_t$ is trained and used in a softmax over $C_t$ classes for the current task (i.e., local $\arg\max$ is used). At inference, consider a new test sample $x$ belonging (in truth) to task $t^*$, the entire head is used: we compute logits for all classes seen so far, and choose the global $\arg\max$. We have: (i) The local $\arg\max$, $\hat{y}_{\text{local}}(x) = \arg\max_{c \in \{1, \ldots, C_{t^*}\}} z_{t^*, c}(x)$, where $z_{t^*, c}(x) = H_t(\mathcal{F}_t(x))$ are the logits restricted to task $t^*$. (ii) The global $\arg\max$, $\hat{y}_{\text{global}}(x) = \arg\max_{(t,c) \in \{1, \ldots, T\} \times \{1, \ldots, C_t\}} z_{t,c}(x)$. So, we will need to ensure a sufficiently high probability that these two predictions coincide, $\Pr(\hat{y}_{\text{local}}(x) = \hat{y}_{\text{global}}(x))$, but it is very difficult to hod in practice due to the diverse nature of streaming tasks. We can show a rough illustrative bound (see Appendix G for more details), $\Pr(\hat{y}_{\text{local}} = \hat{y}_{\text{global}}) \approx \int \left[ \Pr(Z_o \leq z) \right]^M F_{Z^*}(z) \, dz$, where we assume for a task $t^*$, the local maximum logit $Z^*$ has mean $\mu^*$ and variance $\sigma^{*2}$. All out-of-task classes have means $\mu_o < \mu^*$ and variance $\sigma_o^2$, and there are $M$ out-of-task classes in total. $Z_o$ is the logit distribution for a single out-of-task class and $F_{Z^*}$ is the PDF of $Z^*$. If $\mu^*$ is sufficiently larger than $\mu_o$ (and variances are not too large), $Z^*$ will, with high probability, exceed all $M$ out-of-task logits. But as $M$ grows large, this event can become less likely unless the margin $\mu^* - \mu_o$ is also large (which can not hold at streaming task in continual learning). Our experimental results empirically reflect the difficult of this task-agnostic head design.

For the task-specific head at inference, we do not have the loca-argmax-vs-global-argmax issue. Instead, the challenge is to infer the task ID for a testing sample on the fly. In the prior art (Wang et al., 2022d;c; Smith et al., 2023; Wang et al., 2022a), the pretrained model $F_1(\cdot)$ is used as the task-ID query function $q()$, and use $q(x)$ to retrieve the task ID from **the external memory** such as the continually learned task centroids used in (Wang et al., 2022a).

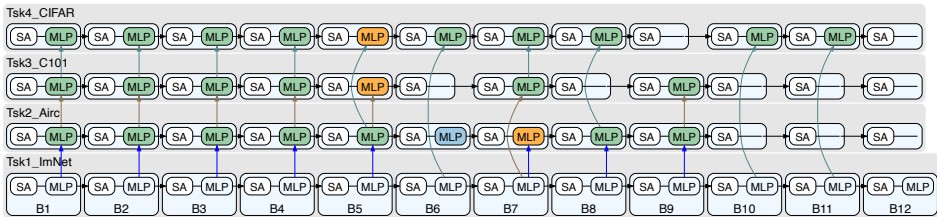

(a) **The MTIL benchmark** (Zheng et al., 2023) consisting of tasks of different nature with `#training images`/`#classes` significantly varying across different tasks.

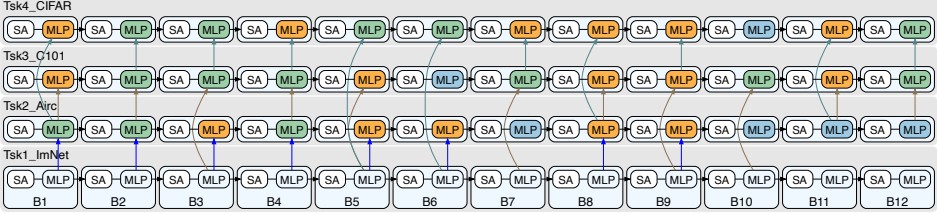

(b) From **ViT-Base trained on Tsk1_ImNet** (with blocks B1 to B12), our CHEEM learns sensible task-tailored models that reflect the task complexity. For example, when learning Caltech 101 (Tsk3_C101), CHEEM learns to `Skip` 5 MLP blocks and Reuse most of the architecture. On the contrary, when learning FGVC Aircraft (Tsk1_Airc), which is a more complex task with larger shift from ImageNet due to its fine-grained nature, CHEEM learns to Adapt the ImageNet parameters in Block 7, adds a New operation in Block 6, and `Skips` the last 3 MLP blocks. See text for details.

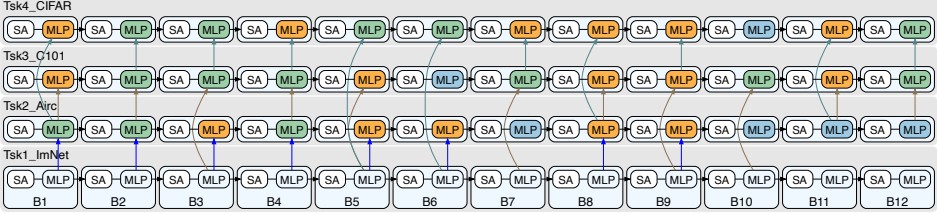

(c) From **DEiT-Tiny trained on Tsk1_ImNet** (with blocks B1 to B12), our CHEEM learns to use multiple Adapt and New operations, without `Skip` operations selected, sensibly different from those with more `Skip` and less New operations learned based on the stronger ViT-Base model.

Figure 2: Examples of CHEEM learning task-tailored models.

In this paper, we choose to focus on learning dynamic backbones in ExfCCL for its better balance between stability and plasticity. We propose a novel formulation for the internal parameter memory, while leveraging the task-specific head design with the external task-centroid memory (Wang et al., 2022a) to eliminate the local-argmax-vs-global-argmax difficulty.

Fig. 1 illustrates our proposed method, **CHEEM** (*Continual Hierarchical-Exploration-Exploitation Memory*), enabling a dynamic learning-to-update ViT backbone that balances stability and plasticity, mitigating catastrophic forgetting through task synergies, in which **a new task learns to automatically reuse/adapt modules from previous similar tasks, to introduce new modules when needed, or to skip some modules when it appears to be an easier task** (see Figs. 2b and 2c). We propose a hierarchical exploration-exploitation (HEE) sampling based neural architecture search (NAS) method for learning the internal memory.

To ensure NAS is computationally efficient, and retain the stability of the backbone to account for tasks in streams that have little training data, we select two components in a ViT block: the down projection layer ($MLP^{Down}$) in the FFN and the projection layer after the MHSA, to be plastic in learning-to-adapt to different tasks using four basic operations:

- `Reuse:` facilitates similar tasks sharing layers for knowledge transfer in continual learning.
- `New:` explores new features for handling tasks that are dissimilar to previous tasks. The New operation enables learning-to-grow the backbone to be skilled at streaming tasks.
- `Adapt:` utilizes Low-Rank Adaptation (LoRA) (Hu et al., 2022), inducing task synergies in ExfCCL in a parameter-efficient way.
- `Skip:` skips the entire MHSA block (when the projection component is used) or the entire FFN block (when the $MLP^{Down}$ is used). It can thus induce much simpler backbones for relatively easier

tasks in a learning-to-shrink manner, especially when a strong backbone such as ViT-Base is used (e.g., from ImageNet to MNIST).

In experiments, to account for the five metrics (average accuracy, average forgetting, average parameter increase and average compute) holistically in ExfCCL, we propose a holistic figure of merits (FoM) based metric to compare CHEEM with baseline methods. Our CHEEM is tested on two challenging benchmarks (MTIL (Zheng et al., 2023) and VDD (Rebuffi et al., 2017)) using both ViT-Base (Dosovitskiy et al., 2021) and DEiT-Tiny (Touvron et al., 2021) and obtains significantly better performance than prompting-based methods (Smith et al., 2023; Wang et al., 2022c;d;a; Tang et al., 2024). Our CHEEM's performance is close to the upper-bound performance using either task-to-task full fine-tuning or task-to-task LoRA based fine-tuning, demonstrating its effectiveness. The learned task-tailored backbones are also sensible, and result in much less overall computing cost across all tasks compared to prompting based methods.

**Our Contributions**. This paper makes three main contributions to the field of ExfCCL using ViT: (i) It poses ExfCCL as a problem of learning two decoupled continual memory in ViT, the external task-centroid memory and the internal parameter memory. (ii) It presents a hierarchical task-synergy exploration-exploitation sampling based NAS method for maintaining the internal memory by learning task-aware dynamic models continually with respect to four operations: Reuse, Adapt, New and Skip, to mitigate catastrophic forgetting. (iii) It shows state-of-the-art performance on two challenging benchmarks (MTIL and VDD) in terms of a proposed figure of merits (FoM) metric, with sensible task-tailored model structures automatically learned.

## 2   OUR PROPOSED CHEEM

This section presents details of our proposed CHEEM. We start with a vanilla $D$-layer ViT model (e.g., the 12-layer ViT-Base) (Dosovitskiy et al., 2021). As illustrated in Fig. 1, we select two components in a Transformer block, the MLP$^{\text{Down}}$ and the project layer after the MHSA to place the internal parameter memory. We provide an ablation study in Appendix H, which empirically supports these two design choices.

### 2.1   THE MIXTURE-OF-EXPERTS REPRESENTATION OF TASK-SYNERGY INTERNAL MEMORY

The proposed internal memory of our CHEEM is represented by a Mixture of Experts (MoEs). Starting with the ViT base model $F_1$, the internal memory at the $l$-th layer in ViT consists of a single expert defined by a tuple,

$$\mathrm{E}_l^{(1,)} = (\theta_l^{(1,)}, \mu_l^1), \quad (1)$$

where the subscript represents the layer index and the list-based superscript shows which task(s) use this expert. $\theta_l^{(1,)}$ are the parameters of the projection layer or the MLP$^{\text{Down}}$ layer and $\mu_l^1 \in R^d$ is the associated mean class-token (CLS) pooled from the training dataset after the model is trained, which is task specific (as indicated by the superscript). For example, if an expert is reused by another task (say, 3) in continual learning, we will have $\mathrm{E}_l^{(1,3,)} = (\theta_l^{(1,3,)}, \mu_l^1, \mu_l^3)$.

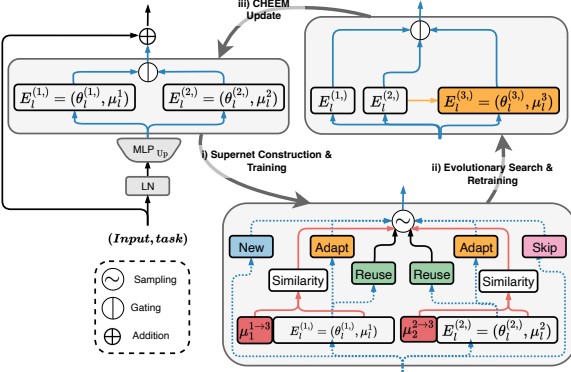

Figure 3: Illustration of CHEEM learning via NAS.

As shown in Fig. 3, for a new task $t$, learning to update CHEEM consists of three components: i) the Supernet construction (the parameter space of updating CHEEM), ii) the Supernet training (the parameter estimation of updating CHEEM), and iii) the target network selection and finetuning (the consolidation of the CHEEM for the task $t$).

### 2.2   SUPERNET CONSTRUCTION VIA Reuse, Adapt, New AND Skip

For clarity, we consider how the space of MoEs of the internal memory is constructed at a single layer $l$ for a new task with CHEEM placed at the MLP$^{\text{Down}}$ (projection) layer, assuming the current memory consists of two experts, $\{\mathrm{E}_l^{(1,)}, \mathrm{E}_l^{(2,)}\}$ (Fig. 3, left). The Supernet is constructed via:

- Reuse: Uses the $\text{MLP}^{\text{Down}}$ (projection) layer from an old task for the new task unchanged, exploiting task synergies during learning.
- Adapt: Introduces a new lightweight LoRA (Hu et al., 2022) component, e.g., $\theta_l^{(3,)} = \theta_l^{(2,)} + B_l \cdot A_l$, where $B_l$ and $A_l$ are low-rank parameter matrices.
- New: Adds a new $\text{MLP}^{\text{Down}}$ (projection) layer, which enables the model to handle corner cases and novel situations.
- Skip: Skips the entire FFN (MHSA) block, which encourages dynamically adjusting the model complexity based on the task complexity.

The bottom of Fig. 3 shows the search space. The Supernet is constructed by reusing and adapting each existing expert at layer $l$, and adding a new and a skip expert. The newly added adapt $(B_l, A_l)$ by LoRA and projection layers will be trained from scratch using the data of a new task only. The right-top of Fig. 3 shows the Adapt operation on top of $\text{E}_l^{(2,)}$ is learned and added, $\text{E}_l^{(3,)} = (\theta_l^{(3,)}, \mu_l^3)$ where $\mu_l^3$ is the mean CLS token pooled for the task 3.

### 2.3 Supernet Training via HEE-NAS

To train the Supernet constructed for a new task $t$, we build on the SPOS method (Guo et al., 2020) due to its efficiency. The basic idea of SPOS is to train a single-path subnetwork from the Supernet by sampling an expert at every layer in each mini-batch of training. One key aspect is the sampling strategy. The vanilla SPOS method uses uniform sampling (i.e., the *pure exploration* (PE) strategy, Fig. 4 top). We propose an exploitation strategy (Fig. 4 bottom), which utilizes a hierarchical sampling method that forms the categorical distribution over the operations in the search space **explicitly based on task synergies computed based on the pooled task-specific CLS tokens**.

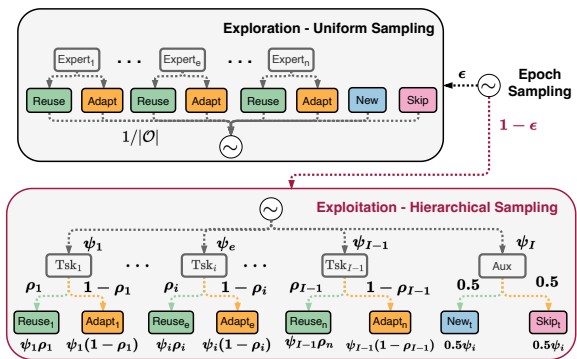

Figure 4: Illustration of the proposed HEE sampling based NAS. It integrates the vanilla exploration strategy (top) and the proposed exploitation strategy (bottom) with an epoch-wise scheduling.

Consider a new task $t$ with the training dataset $D_t^{train}$, with the current supernet consisting of $t-1$ task-specific target networks, we first run inference of the $t-1$ target networks on $D_t^{train}$ to pool initial CLS tokens for each expert, e.g., $\mu_l^{1\to3}$ and $\mu_l^{2\to3}$ in the bottom of Fig. 3. Consider one expert $\text{E}_l^{(i,j,)}$ at the $l$-th layer which is shared by two previous tasks $i$ and $j$ with their mean CLS tokens $\mu_l^i$ and $\mu_l^j$ respectively, we have the pooled CLS tokens for the current task $t$, $\mu_l^{i\to t}$ and $\mu_l^{j\to t}$, computed accordingly. The task similarity is computed by,

$$S_l^{i,t} = \texttt{NormCosine}(\mu_l^i, \mu_l^{i\to t}), \qquad (2)$$

where $\texttt{NormCosine}(\cdot, \cdot)$ is the Normalized Cosine Similarity, which is calculated by scaling the Cosine Similarity score between $-1$ and $1$ using the minimum and the maximum Cosine Similarity scores from all the experts in all the MHSA blocks of the ViT. This normalization is necessary to increase the difference in magnitudes of the similarities between tasks, which results in better Expert sampling distributions during the sampling process in our experiments. The task similarity score will be used in sampling the Reuse and Adapt operations.

For the new task $t$, we also have the New expert and the Skip expert at each layer $l$, for which we do not have similarity scores. Instead, we introduce an auxiliary expert, Aux (see the bottom of Fig. 4) which gives equally-likely chance to select the New expert or the Skip expert once sampled in NAS. For the Aux expert itself, the similarity score between it and the new task $t$ is specified by,

$$S_l^{aux,t} = -\max_{i=1}^{t-1} S_l^{i,t}, \qquad (3)$$

which intuitively means we probabilistically resort to the New operation or the Skip operation when other experts turn out not "helpful" for the task $t$.

At each layer $l$ in the ViT, for a new task $t$, the task-similarity oriented operation sampling is realized by a 2-level hierarchical sampling, as illustrated in the right of Fig. 4:

- The first level uses a categorical distribution with the maximum number of entries being $t$ consisting of at most the previous $t-1$ tasks (some of which may use `Skip` and thus will be ignored) and the `Aux` expert. The categorical distribution $(\psi_1, \cdots, \psi_i, \cdots, \psi_{I-1}, \psi_I)$ is computed by the Softmax function over the similarity scores defined above, where $I \leq t$.
- With a previous task $i$ sampled with the probability $\psi_i$, at the second level of sampling, we sample the `Reuse` operation for the associated expert using a Bernoulli distribution with the succcess rate computed by the Sigmoid function of the task similarity score defined by $\rho_i = \frac{1}{1+\exp(-S_l^{i,t})}$, and the `Adapt` operation with probability $1 - \rho_i$.

### 2.4 Compute-Aware Target Network Selection

After the Supernet is trained, we propose a compute-sensitive evolutionary search on top of (Real et al., 2019). It first draws a population with a predefined number of candidate architectures from the trained Supernet using our proposed HEE sampling method. It then "evolves" the population via the crossover and the mutation operations. At each "evolving" iteration, the population is evaluated and sorted based on the trade-off between the validation performance and the compute of candidates: we predefine a performance tolerance threshold $\tau$ (e.g., $\tau = 2\%$) to group candidate networks, and rank candidate networks in each group based on their compute in the increasing order. With the top-$k$ candidates after evaluation and sorting (the number $k$ is predefined), for crossover, two randomly sampled candidate networks in the top-$k$ are crossed to produce a new target network. For mutation, a randomly selected candidate in the top-$k$ mutates its every choice block with probability (e.g., 0.1) to produce a new candidate. Crossover and mutation are repeated to generate sufficient new candidate target networks to form the population for the next "evolving" iteration. We study the effect of varying the $\tau$ in Figure 9b in Appendix F.

### 2.5 Target Network Retraining From Scratch

After the target network for a new task is selected, we retrain the newly added layers by the `New` and `Adapt` operations from scratch (random initialization), rather than keeping or warming-up from the weights from the Supernet training. This is based on the observations in network pruning that it is the neural architecture topology that matters and that the warm-up weights may not need to be preserved to ensure good performance on the target dataset (Liu et al., 2019b). Our experiments during the development of CHEEM confirms this observation.

### 2.6 Balancing Exploration and Exploitation

As illustrated in Fig. 4, to harness the best of the pure exploration strategy and the proposed exploitation strategy, we apply epoch-wise exploration and exploitation sampling for simplicity. For the pure exploration, we directly uniformly sample the experts at a layer $l$, consisting of the $n$ experts from the previous $t-1$ tasks, and the `New` and `Skip` operations, where $n \leq t-1$. At the beginning of an epoch in the Supernet training, we choose the pure exploration strategy with a probability of $\epsilon_1$ (e.g., 0.3), and the hierarchical sampling strategy with a probability of $1 - \epsilon_1$. Similarly, when generating the initial population during the evolutionary search, we draw a candidate target network from a uniform distribution over the operations with a probability of $\epsilon_2$, and from the hierarchical sampling process with a probability of $1 - \epsilon_2$, respectively. In practice, we set $\epsilon_2 > \epsilon_1$ (e.g., $\epsilon_2 = 0.5$) to encourage more exploration during the evolutionary search, while encouraging more exploitation for faster learning in the Supernet training. We study the effect of $\epsilon_1$ and $\epsilon_2$ in Figure 9a in Appendix F.

## 3 Experiments

**Data.** We evaluate CHEEM on two challenging benchmarks, the MTIL benchmark (Zheng et al., 2023) and the VDD benchmark (Rebuffi et al., 2017), both consisting of tasks from varying domains with different complexities. While MTIL is largely balanced in terms of classes per task, VDD presents a much larger class imbalance. For example, out of the total 2128 classes (excluding ImageNet-1k), Omniglot contains 1623 classes, whereas DTD contains only 47. Further details of the benchmarks can be found in Appendix E.

**Pretrained Models in ExfCCL.** We test two settings: one strong ViT-Base pretrained on the ImageNet-21k and fine-tuned on the ImageNet-1k, and the other relatively weaker DEiT-Tiny

Table 1: Comparison of Average Accuracy and Forgetting **on the MTIL benchmark** with three different seeds. Accuracy of individual tasks are in the supplementary (Table 9).

| Method | ViT-Base | | | DEiT-Tiny | | |
|---|---|---|---|---|---|---|
| | Avg. Acc | Avg. Frgt. | FLOPs | Avg. Acc | Avg. Frgt. | FLOPs |
| Full Finetuning | $88.1 \pm 0.0$ | - | - | $75.3 \pm 0.1$ | - | - |
| LoRA Finetuning | $87.4 \pm 0.0$ | - | - | $74.6 \pm 0.1$ | - | - |
| CHEEM (MLP$^{Down}$) | $\mathbf{85.9} \pm 0.3$ | $1.7 \pm 0.1$ | 62.3 | $\mathbf{74.5} \pm 0.3$ | $1.9 \pm 0.0$ | 4.5 |
| EWC | $44.6 \pm 6.4$ | $23.8 \pm 6.5$ | 33.7 | $35.3 \pm 0.3$ | $7.3 \pm 0.6$ | 2.2 |
| CODA-Prompt | $40.2 \pm 1.2$ | $25.3 \pm 1.8$ | 70.3 | $5.6 \pm 0.3$ | $42.6 \pm 0.8$ | 5.0 |
| DualPrompt | $33.8 \pm 0.4$ | $22.1 \pm 0.4$ | 70.3 | $30.9 \pm 0.3$ | $17.5 \pm 0.3$ | 5.0 |
| L2P | $26.6 \pm 0.2$ | $31.0 \pm 0.3$ | 70.3 | $23.2 \pm 0.1$ | $25.8 \pm 0.4$ | 5.1 |
| S-Prompts | $81.6 \pm 0.4$ | $1.6 \pm 0.1$ | 67.6 | $67.3 \pm 0.4$ | $1.8 \pm 0.1$ | 4.4 |
| DIKI | $76.4 \pm 0.0$ | $2.0 \pm 0.0$ | 42.5 | $67.6 \pm 0.1$ | $1.8 \pm 0.0$ | 2.8 |
| LoRA (MLP$^{Down}$) | $84.7 \pm 0.0$ | $1.6 \pm 0.1$ | 68.2 | $71.1 \pm 0.0$ | $1.9 \pm 0.0$ | 4.5 |

Table 2: FoM (Eqn. 6) of CHEEM (MLP$^{Down}$) against baselines **on the MTIL benchmark**.

| Method | ViT-Base | DEiT-Tiny |
|---|---|---|
| EWC | 10.5 | 26.0 |
| CODA-Prompt | 24.2 | 84.6 |
| Dual-Prompt | 27.4 | 73.7 |
| L2P | 31.1 | 79.8 |
| S-Prompts | 3.2 | 10.5 |
| DIKI | 3.6 | 6.4 |
| LoRA (MLP$^{Down}$) | 1.7 | 5.7 |

Table 3: Comparison of Average Accuracy and Forgetting **on the VDD benchmark** with three different seeds. Accuracy of individual tasks are in the Appendix (Table 10).

| Method | ViT-Base | | | DEiT-Tiny | | |
|---|---|---|---|---|---|---|
| | Avg. Acc | Avg. Frgt. | FLOPs | Avg. Acc | Avg. Frgt. | FLOPs |
| Full Finetuning | $88.7 \pm 0.1$ | - | - | $76.21 \pm 0.1$ | - | - |
| LoRA Finetuning | $86.8 \pm 0.1$ | - | - | $76.3 \pm 0.3$ | - | - |
| CHEEM (MLP$^{Down}$) | $\mathbf{86.7} \pm 0.2$ | $0.4 \pm 0.0$ | 61.6 | $\mathbf{76.18} \pm 0.1$ | $1.0 \pm 0.0$ | 4.5 |
| EWC | $44.0 \pm 1.3$ | $5.1 \pm 1.1$ | 33.7 | $33.7 \pm 0.2$ | $1.5 \pm 0.1$ | 2.2 |
| CODA-Prompt | $24.9 \pm 2.2$ | $26.1 \pm 0.8$ | 70.3 | $1.1 \pm 0.1$ | $37.6 \pm 0.4$ | 5.0 |
| DualPrompt | $28.1 \pm 0.4$ | $3.2 \pm 0.5$ | 70.3 | $19.4 \pm 0.6$ | $10.5 \pm 0.5$ | 5.0 |
| L2P | $23.9 \pm 0.7$ | $9.0 \pm 0.6$ | 70.3 | $11.5 \pm 0.8$ | $20.9 \pm 1.7$ | 5.1 |
| S-Prompts | $78.6 \pm 0.1$ | $0.4 \pm 0.0$ | 67.6 | $65.8 \pm 0.3$ | $0.9 \pm 0.0$ | 4.4 |
| DIKI | $65.9 \pm 0.1$ | $0.1 \pm 0.0$ | 42.5 | $58.3 \pm 0.1$ | $0.6 \pm 0.0$ | 2.8 |
| LoRA (MLP$^{Down}$) | $86.0 \pm 0.1$ | $0.3 \pm 0.0$ | 68.2 | $74.0 \pm 0.3$ | $1.1 \pm 0.0$ | 4.5 |

Table 4: FoM (Eqn. 6) of CHEEM (MLP$^{Down}$) against baselines **on the VDD benchmark**.

| Method | ViT-Base | DEiT-Tiny |
|---|---|---|
| EWC | 12.1 | 678.9 |
| CODA-Prompt | 35.9 | 2811.3 |
| Dual-Prompt | 34.1 | 2130.6 |
| L2P | 36.4 | 2431.6 |
| S-Prompts | 5.5 | 338.8 |
| DIKI | 7.8 | 370.8 |
| LoRA (MLP$^{Down}$) | 1.5 | 73.6 |

trained on ImageNet-1k. The overall objective is two-fold: (i) to observe how different continual learning methods cope with different pretrained conditions, and (ii) to verify if our CHEEM can adaptively learn sensible task-tailored models, e.g., learning more `Skip` (`New`) when the strong (weak) pretrained ViT is used.

**Implementation Details**: In all our experiments, we apply CHEEM to the MLP$^{Down}$ layer, unless stated otherwise. We provide further implementation details in Appendix E.

**Baselines.** We compare with four types of baselines:

- The classic *Elastic Weight Consolidation* (EWC) (Kirkpatrick et al., 2017a) method.
- *State-of-the-art prompting based methods*: CODA-Prompt (Smith et al., 2023), Dual-Prompt (Wang et al., 2022c), Learning-to-Prompt (L2P) (Wang et al., 2022d), S-Prompts (Wang et al., 2022a) and DIKI (Tang et al., 2024).
- *Parameter-Efficient Fine-Tuning (PEFT) based continual learning*: we test LoRA (Hu et al., 2022) as the alternative internal parameter memory while using the same external task-centroid memory as our CHEEM. This is a special case of our CHEEM, using the LoRA `Adapt` operation and without NAS.
- *Upper-bound task-to-task fine-tuning*: we test two settings, full task-to-task fine-tuning, and LoRA (Hu et al., 2022) based task-to-task PEFT, from the pretrained model to each task individually.

**Metrics**: We measure the performance of CHEEM using three metrics: Average Accuracy, Average Forgetting (Chaudhry et al., 2018), and Figure of Merit. Let $(\mathcal{F}_i, \mathcal{H}_i)$ be the feature backbone and the classifier heads after completion of task $i$, and $a_{i,j} = Acc(D_j^{test}; \mathcal{F}_i, \mathcal{H}_i)$ be the Top-1 accuracy on the testing data for task $j$ computed using $(\mathcal{F}_i, \mathcal{H}_i)$. The Average Accuracy ($A\mathbb{A}$) after $\mathcal{T} \setminus \{T_1\}$ and Average Forgetting ($A\mathbb{F}$) after $\mathcal{T} \setminus \{T_1, T_N\}$ tasks are defined as

$$A\mathbb{A} = \frac{1}{N-1} \sum_{t=2}^{N} \text{Acc}(D_t^{test}; \mathcal{F}_N, \mathcal{H}_N), \quad (4) \qquad A\mathbb{F} = \frac{1}{N-2} \sum_{t=2}^{N-1} \left( \max_{j \in [t,N]} a_{j,t} - a_{N,t} \right), \quad (5)$$

We propose a new pair-wise metric, **Figure of Merit (FoM)**, to explicitly and holistically compare two methods (e.g., our CHEEM against another baseline) with respect to their respective average accuracies and model complexities, where the model complexity is measured using FLOPs. For two methods $m$ and $n$, we define the FoM as

$$\text{FoM}(m, n) = \frac{A\mathbb{A}^{UpperBound} - A\mathbb{A}^n}{A\mathbb{A}^{UpperBound} - A\mathbb{A}^m} \cdot \frac{\text{FLOPs}^n}{\text{FLOPs}^m}, \qquad (6)$$

Table 5: Task-wise FLOPs for the MTIL and VDD benchmarks using ViT-Base as pretrained model.

| MTIL | | | | | | VDD | | | | |
|---|---|---|---|---|---|---|---|---|---|---|
| **Airc** | **C101** | **CIFAR** | **DTD** | **ESAT** | **Flwr** | **CIFAR** | **DPed** | **OGlt** | **SVHN** | **UCF** |
| $62.8 \pm 0.9$ | $59.0 \pm 0.9$ | $64.0 \pm 0.0$ | $64.6 \pm 0.9$ | $57.8 \pm 0.9$ | $62.1 \pm 0.1$ | $63.5 \pm 1.7$ | $54.1 \pm 0.9$ | $62.2 \pm 1.5$ | $56.7 \pm 0.0$ | $65.5 \pm 1.8$ |
| **F101** | **MNIST** | **Pets** | **Cars** | **SUN397** | **Avg** | **GTSR** | **Flwr** | **Airc** | **DTD** | **Avg** |
| $65.2 \pm 1.7$ | $56.0 \pm 0.9$ | $62.8 \pm 0.9$ | $65.9 \pm 1.5$ | $65.2 \pm 0.9$ | $62.3 \pm 0.3$ | $57.8 \pm 1.7$ | $62.2 \pm 0.0$ | $66.1 \pm 0.1$ | $66.6 \pm 0.9$ | $61.6 \pm 0.3$ |

where $A\mathbb{A}^{\text{UpperBound}}$ represents the average accuracy of upper-bound full task-to-task fine-tuning, and FLOPs is the computing cost. If a method $m$ has smaller performance gap against the upper bound and smaller computing cost than another method $n$, FoM$(m, n)$ will be greater than 1. There is a trade-off between the first performance ratio and the second cost ratio. Intuitively, FoM$(m, n)$ represents the relative magnitude of method $m$ being better than $n$.

## 3.1 PERFORMANCE COMPARISONS ON MTIL AND VDD

Table 1 and Table 3 show the Average Accuracy, Average Forgetting, and the runtime FLOPs on MTIL and VDD benchmarks respectively. **Our CHEEM outperforms all the baseline methods by large margins.**

Table 2 and Table 4 show the FoM of CHEEM on MTIL and VDD benchmarks respectively. **The FoM shows that CHEEM can balance Average Accuracy and FLOPs**, whereas the baseline methods fall short on either of both of the axes. For example, on both MTIL and VDD, DIKI achieves lower FLOPs, but sacrifices performance. LoRA (MLP$^{down}$) achieves Average Accuracy close to CHEEM, but requires higher FLOPs as it cannot skip modules. The FoM of CHEEM against baselines for DEiT-Tiny are significantly large on VDD since CHEEM almost reaches the full fine-tuning performance (76.18% vs 76.21%), resulting in a very large accuracy gap ratio term in Eqn. 6.

**Our CHEEM closely approaches full fine-tuning performance**. On MTIL, CHEEM achieves 85.9% vs 88.1% for ViT-Base, and 74.5% vs 75.3% for DEiT-Tiny. On VDD, CHEEM achieves 86.7% vs 88.7% for ViT-Base, and 76.2% vs 76.2% for DEiT-Tiny.

**Importance of structure updates to backbone - CHEEM vs Prompting-based Baselines**: Three prompting-based methods (CODA-Prompt, DualPrompt and L2P) perform even worse than EWC for both ViT-Base and DEiT-Tiny, mainly due to the aforementioned discrepancy between global-argmax-vs-local-argmax in their head classifier designs. CODA-Prompt almost completely failed for DEiT-Tiny with 5.62% average accuracy. S-Prompts works the best among prompting based methods, but is still inferior to our CHEEM: 4% drop (81.62% vs 85.88%) for ViT-Base, and 7% drop (67.33% vs 74.51%) for DEiT-Tiny. This shows the importance of inferring task IDs on-the-fly for streaming tasks with significantly varying distributions of classes. Overall, **the superior performance by our CHEEM shows the importance of structurally and dynamically updating the backbone with the task-synergy internal memory**.

**Importance of Search - CHEEM vs LoRA**: Both are applied to MLP$^{\text{Down}}$ and use the same external task-centroid memory for task IDs inference. The LoRA counterpart is a special case of our CHEEM (only learning `Adapt` operation without NAS). The improvement by CHEEM, 1% increase for ViT-Base and 3.45% increase for DEiT-Tiny show the benefits of HEE-NAS, especially for weaker backbone such as DEiT-Tiny, leading to more competent ExfCCL that is less sensitive to starting feature backbone. We note that the LoRA counterpart outperforms all prompting based baselines, showing the importance of introducing new model parameters in ExfCCL.

**CHEEM vs EWC.** EWC suffers from catastrophic forgetting due to the restriction of maintaining a single shared backbone, and can only reach average accuracy 44.58% for ViT-Base and 35.33% for DEiT-Tiny.

## 3.2 CHEEM LEARNS SENSIBLE AND TASK-TAILORED MODEL STRUCTURES

Intuitively, easier tasks should require lesser FLOPs in continual learning. Table 5 shows that CHEEM allocates lower FLOPs to easier tasks like MNIST and ESAT. Figs. 2b and 2c show some examples of architectures learned by CHEEM on the MTIL benchmark. **These sensible model structures are unique to our CHEEM in comparisons to other baselines.** They also show interesting yet "irregular" model configurations caused by learned `Skip` operations in different blocks in Fig. 2b: two consecutive Transformer blocks with one block comprising only the attention component (for token mixing) without the FFN (for channel mixing). Fig. 5 in the Appendix shows the sensible model structures learned by CHEEM on VDD.

Table 6: Comparisons of two selected CHEEM placements in a ViT block.

| Dataset | Method | ViT-Base | | | DEiT-Tiny | | |
|---|---|---|---|---|---|---|---|
| | | Avg. Acc | FLOPs | %Param | Avg. Acc | FLOPs | %Param |
| MTIL | MLP$^{Down}$ | **85.88** | 62.31 | 0.40 | **74.51** | 4.47 | 6.32 |
| | Attn Proj | 85.57 | 65.91 | 0.25 | 74.39 | 4.42 | 1.79 |
| VDD | MLP$^{Down}$ | 86.71 | 61.63 | 1.45 | **76.18** | 4.49 | 5.98 |
| | Attn Proj | 87.23 | 65.91 | 0.27 | 75.97 | 4.68 | 2.10 |

Table 7: Comparisons of our HEE and Uniform (i.e., Pure Exploration) sampling during Supernet NAS training on the MTIL benchmark.

| Method | ViT-Base | | | DEiT-Tiny | | |
|---|---|---|---|---|---|---|
| | Avg. Acc | FLOPs | %Param | Avg. Acc | FLOPs | %Param |
| HEE | **85.88** | 62.31 | 0.25 | 74.51 | 4.47 | 6.32 |
| Uniform | 84.74 | 61.82 | 5.89 | **75.05** | 4.47 | 14.93 |

## 3.3 ABLATION STUDIES

**CHEEM Placement: MLP$^{Down}$ vs. Projection** Table 6 shows the comparisons. Both of the placements of CHEEM achieve on-par average accuracy performance. However, due to the size of the FFN layers, when skipping the FFN block rather than the attention block, CHEEM (MLP$^{Down}$) shows better FLOPs reduction.

**Sampling in NAS: HEE vs. Uniform** Table 7 shows the comparisons. Although both methods achieve on-par average accuracy, the uniform sampling method leads to much higher number of new parameter increase (due to `New` and `Adapt`): 5.89% vs 0.25% for ViT-Base (as seen in Figure 7 in the supplementary text), and 14.93% vs 6.32% for DEiT-Tiny. The promising performance of uniform sampling based NAS shows the representational power of our proposed internal parameter memory using the four basic operations (`Reuse, Adapt, New and Skip`). The parsimoniousness of HEE-NAS hightlights its efficacy in continual learning by effectively leveraging task synergies, especially towards many more tasks in streams beyond the MTIL and VDD benchmarks.

## 4 RELATED WORK

For exemplar-free continual learning, *Regularization Based approaches* explicitly control the plasticity of the model by preventing the parameters of the model from deviating too far from their stable values learned on the previous tasks when learning a new task (Kirkpatrick et al., 2017a; Aljundi et al., 2018; 2019; Douillard et al., 2020; Nguyen et al., 2018; Kirkpatrick et al., 2017b; Li & Hoiem, 2018; Zenke et al., 2017; Schwarz et al., 2018). These approaches aim to balance the stability and plasticity of a fixed-capacity model. *Dynamic Models* aim to use different parameters for each task to eliminate the use of stored exemplars. Dynamically Expandable Network (Yoon et al., 2018) adds neurons to a network based on learned sparsity constraints and heuristic loss thresholds. PathNet (Fernando et al., 2017) finds task-specific submodules from a dense network, and only trains submodules not used by other tasks. Progressive Neural Networks (Rusu et al., 2016) learn a new network per task and adds lateral connections to the previous tasks' networks. (Rebuffi et al., 2017) learns residual adapters which are added between the convolutional and batch normalization layers. (Aljundi et al., 2017) learns an expert network per task by transferring the expert network from the most related previous task. The L2G (Li et al., 2019) uses Differentiable Architecture Search (DARTS) (Liu et al., 2019a) to determine if a layer can be reused, adapted, or renewed for a task, which is tested for ConvNets and the learning-to-grow operations are applied uniformly at each layer in a ConvNet. Our method is motivated by the L2G method, but with substantially significant differences.

Recently, there has been increasing interest in continual learning using Vision Transformers (Wang et al., 2022d;c; Xue et al., 2022; Ermis et al., 2022; Douillard et al., 2022; Pelosin et al., 2022; Yu et al., 2021; Li et al., 2022a; Iscen et al., 2022; Wang et al., 2022a;b; Mohamed et al., 2023; Gao et al., 2023). *Prompt Based approaches* learn external parameters appended to the data tokens that encode task-specific information useful for classification (Wang et al., 2022d;a; Douillard et al., 2022; Smith et al., 2023; Wang et al., 2022c; Tang et al., 2024). Our proposed method is complementary to prompting-based methods.

## 5 CONCLUSION

We present a method of transforming Vision Transformers (ViTs) for exemplar-free class-incremental continual learning (ExfCCL), dubbed **CHEEM** (Continual Hierarchical-Exploration-Exploitation Memory). Our CHEEM consists of the external (task-centroid) memory and the internal (parameter) memory. The former is for task ID inference for test data based on clustered task centroids in training. The latter is realized by a proposed Hierarchical-Exploration-Exploitation (HEE) sampling based neural architecture search algorithm. The external and internal memory are maintained in a decoupled way. Our CHEEM is tested on two challenging benchmarks, the MTIL and VDD benchmarks. It obtains state-of-the-art performance on both benchmarks, outperforming the prior art by a large margin, with sensible CHEEM structures continually learned.

## REPRODUCIBILITY STATEMENT

All the hyperparameter and coding framework details required to reproduce our experiments are present in Appendix E. We will release the full code upon acceptance.

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

# A  EXAMPLES OF CHEEM CONTINUALLY LEARNED ON THE VDD BENCHMARK

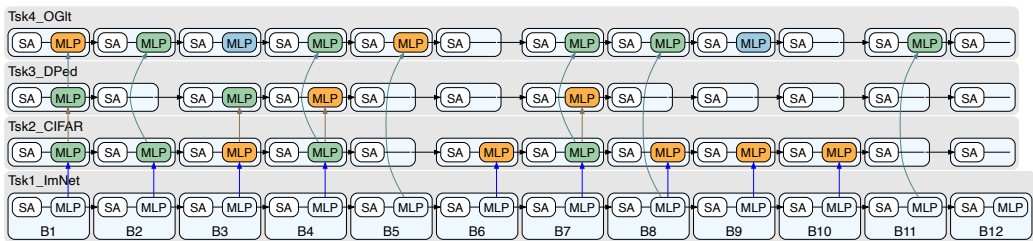

(a) **The VDD benchmark** Rebuffi et al. (2017) consisting of tasks of different nature with `#training images/#classes` significantly varying across different tasks.

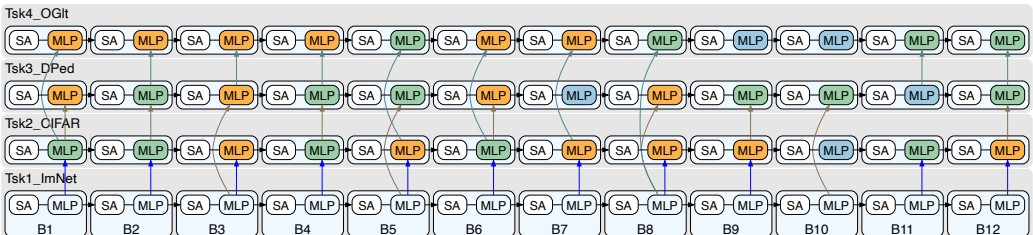

(b) From **ViT-Base trained on Tsk1_ImNet** (with blocks B1 to B12), our CHEEM learns sensible task-tailored models that reflect the task complexity. For example, when learning Daimer Pedestrian Classification (Tsk3_DPed), CHEEM learns to `Skip` 8 MLP blocks and Reuse most of the architecture. When learning Omniglot (Tsk3_Oglt), which has a larger shift from ImageNet, CHEEM learns to Adapt the ImageNet parameters in Blocks 1 and 5, adds New operations in Blocks 3 and 9, and `Skips` blocks 6, 10 and 12.

(c) From **DEiT-Tiny trained on Tsk1_ImNet** (with blocks B1 to B12), our CHEEM learns to use multiple Adapt and New operations, without `Skip` operations selected, sensibly different from those with more `Skip` and less New operations learned based on the stronger ViT-Base model.

Figure 5: Examples of CHEEM learning task-tailored models.

## B EFFECTS OF STREAMING TASK ORDERS

We verify the effect of different task orders on the performance of CHEEM. Table 8 shows that CHEEM is robust to task orders on the MTIL benchmark.

Table 8: Results of learning CHEEM on the MTIL benchmark with three different streaming task orders.

| SUN | Airc | DTD | F101 | Cars | C101 | CIFAR | ESAT | Flwr | MNIST | Pets | Avg. Acc. | Avg. Frgt. |
|------|------|------|------|------|------|------|------|------|------|------|------|------|
| 68.59 | 67.87 | 69.15 | 89.02 | 83.60 | 84.41 | 90.47 | 98.56 | 97.82 | 99.65 | 93.05 | 85.65 | 1.38 |
| **C101** | **CIFAR** | **ESAT** | **Flwr** | **MNIST** | **Pets** | **DTD** | **Cars** | **F101** | **Airc** | **SUN** | **Avg. Acc.** | **Avg. Frgt.** |
| 78.23 | 90.36 | 98.42 | 97.76 | 99.66 | 91.77 | 69.15 | 84.48 | 89.30 | 66.13 | 66.86 | 84.74 | 2.47 |
| **MNIST** | **SUN** | **Flwr** | **DTD** | **C101** | **Cars** | **Pets** | **F101** | **CIFAR** | **Airc** | **ESAT** | **Avg. Acc.** | **Avg. Frgt.** |
| 99.63 | 68.58 | 98.11 | 67.87 | 84.52 | 84.39 | 92.53 | 88.50 | 90.88 | 69.10 | 97.78 | 85.63 | 1.28 |

## C FULL LEARNED CHEEM ON MTIL

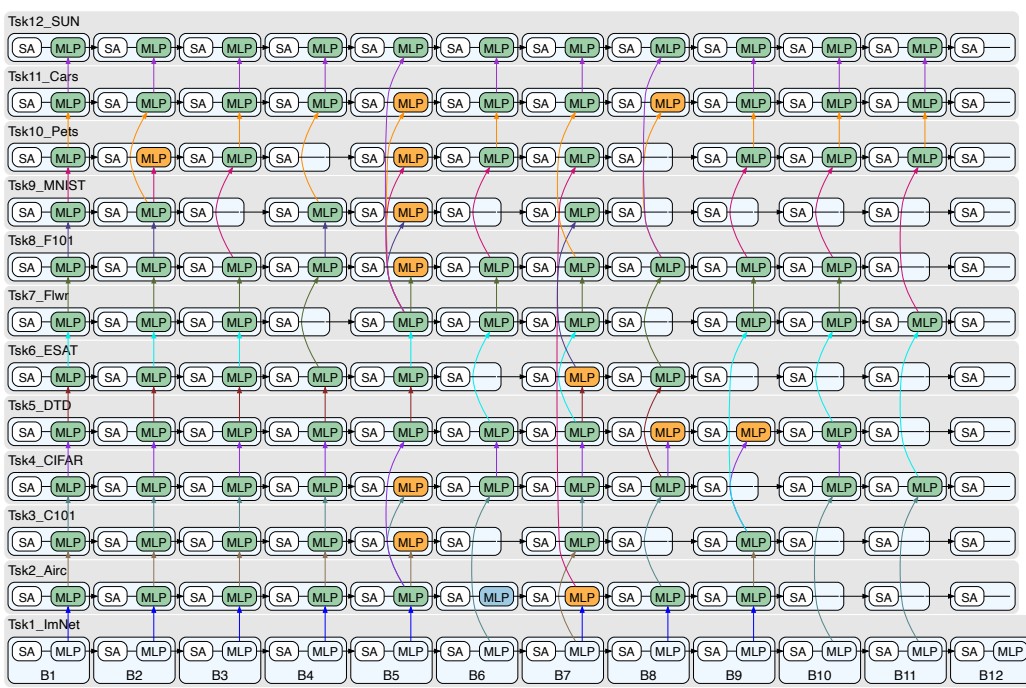

Figure 6: Figure 2(b) from the main text reproduced on the full benchmark. From **ViT-Base trained on Tsk1_ImNet** (with blocks B1 to B12), our CHEEM learns sensible task-tailored models that reflect the task complexity. For example, when learning Caltech 101 (Tsk3_C101), CHEEM learns to Skip 5 MLP blocks and Reuse most of the architecture. On the contrary, when learning FGVC Aircraft (Tsk1_Airc), which is a more complex task with larger shift from ImageNet due to its fine-grained nature, CHEEM learns to Adapt the ImageNet parameters in Block 7, adds a New operation in Block 6, and Skips the last 3 MLP blocks. When learning MNIST, CHEEM skips 8 MLP blocks, accounting for the easy nature of the task.

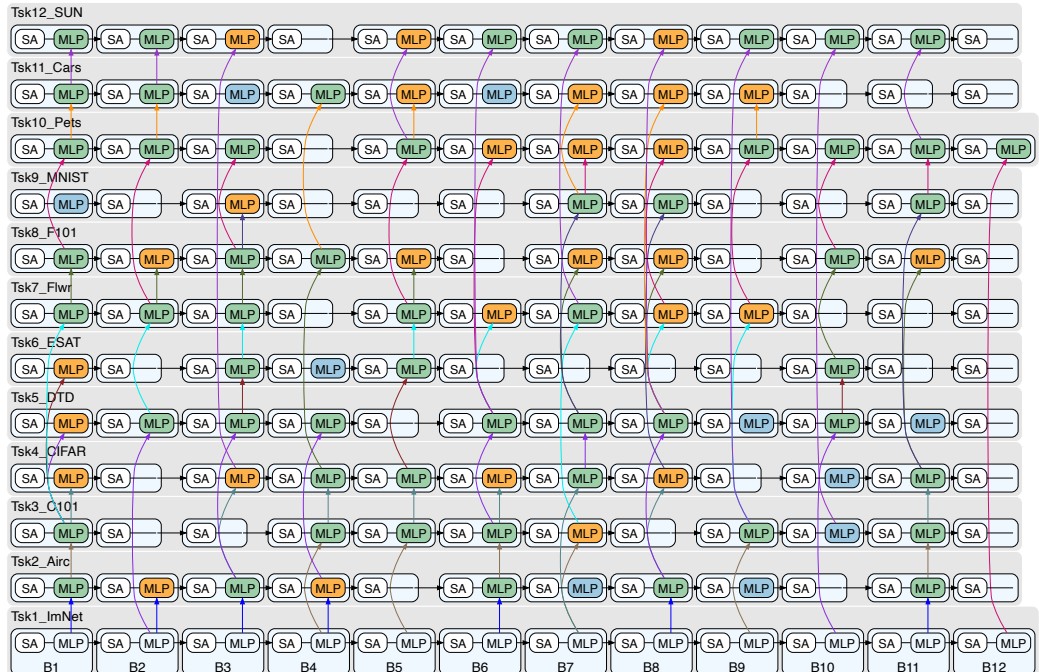

Figure 7: **ViT-Base trained on Tsk1_ImNet** (with blocks B1 to B12), with **Pure Exploration in CHEEM**. While pure exploration accounts for task complexity through the skip operation, it also adds more many more Adapt and New operations as compared to the proposed Hierarchical Exploration-Exploitation scheme (Figure 6). This shows that the HEE sampling scheme can effectively leverage task synergies and reuse previous parameter memories.

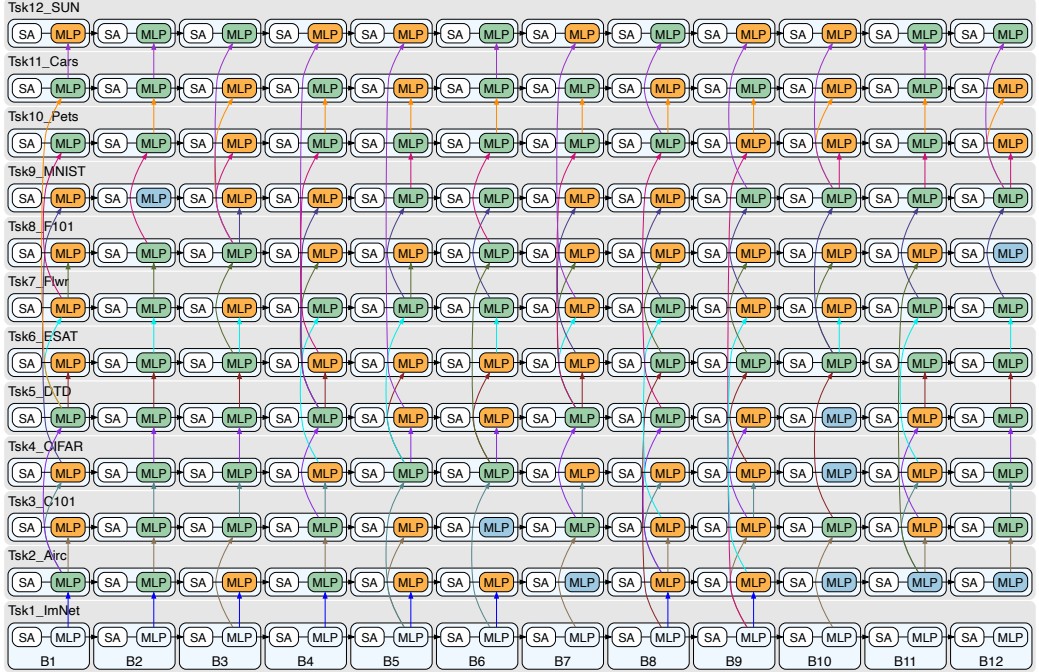

Figure 8: Figure 2(c) reproduced on the full benchmark. From **DEiT-Tiny trained on Tsk1_ImNet** (with blocks B1 to B12), our CHEEM learns to use multiple Adapt and New operations, without Skip operations selected, sensibly different from those with more Skip and less New operations learned based on the stronger ViT-Base model.

# D   FULL RESULTS

Table 9: **MTIL**: Full results on the MTIL benchmark, extending Table 1 in the main text.

| Method | Airc | C101 | CIFAR | DTD | ESAT | Flwr | F101 | MNIST | Pets | Cars | SUN | Avg. Acc | Avg. Frgt. |
|---|---|---|---|---|---|---|---|---|---|---|---|---|---|
| **ViT Base** | | | | | | | | | | | | | |
| Full Finetuning | 69.87 | 98.32 | 90.66 | 77.46 | 98.78 | 97.87 | 88.46 | 99.70 | 92.85 | 85.42 | 69.89 | 88.12 ± 0.04 | - |
| LoRA Finetuning | 63.86 | 97.77 | 91.35 | 77.59 | 98.84 | 98.83 | 88.41 | 99.69 | 93.14 | 80.26 | 71.96 | 87.43 ± 0.01 | - |
| CHEEM ($MLP^{Down}$, HEE) | 69.77 | 84.86 | 90.27 | 68.48 | 98.31 | 97.54 | 89.48 | 99.60 | 92.88 | 84.94 | 68.58 | 85.88 ± 0.29 | 1.73 ± 0.05 |
| CHEEM ($MLP^{Down}$, PE) | 69.97 | 84.96 | 90.21 | 66.74 | 97.97 | 97.32 | 86.97 | 99.50 | 92.32 | 82.11 | 64.06 | 84.74 ± 0.26 | 1.72 ± 0.05 |
| CHEEM (Attn Proj, HEE) | 69.92 | 83.00 | 90.44 | 66.54 | 98.31 | 97.53 | 88.94 | 99.60 | 92.90 | 85.50 | 68.62 | 85.57 ± 0.27 | 1.67 ± 0.03 |
| EWC | 39.10 | 40.90 | 43.93 | 12.98 | 61.43 | 22.24 | 51.81 | 96.20 | 60.65 | 12.64 | 48.46 | 44.58 ± 6.35 | 23.80 ± 6.53 |
| CODA-Prompt | 0.91 | 19.14 | 75.60 | 7.39 | 38.26 | 24.40 | 84.62 | 97.32 | 36.02 | 12.61 | 46.17 | 40.22 ± 1.22 | 25.25 ± 1.78 |
| DualPrompt | 3.08 | 14.40 | | 3.48 | 46.45 | 6.00 | 85.46 | 68.43 | 24.13 | 5.88 | 30.78 | 33.82 ± 0.35 | 22.11 ± 0.42 |
| L2P | 1.22 | 17.35 | 78.81 | 3.46 | 30.39 | 4.67 | 78.47 | 16.83 | 23.45 | 4.62 | 33.40 | 26.61 ± 0.16 | 30.96 ± 0.27 |
| S-Prompts | 53.78 | 82.54 | 88.26 | 65.44 | 96.71 | 98.51 | 84.64 | 99.23 | 92.88 | 70.09 | 65.79 | 81.62 ± 0.35 | 1.64 ± 0.05 |
| DIKI | 52.29 | 91.68 | 89.10 | 63.95 | 96.31 | 30.22 | 86.55 | 98.37 | 92.24 | 70.22 | 69.74 | 76.42 ± 0.04 | 1.96 ± 0.02 |
| LoRA ($MLP^{Down}$) | 63.78 | 85.68 | 90.52 | 67.98 | 98.41 | 98.51 | 87.26 | 99.69 | 92.51 | 79.79 | 67.59 | 84.70 ± 0.01 | 1.64 ± 0.11 |
| **DiET Tiny** | | | | | | | | | | | | | |
| Full Finetuning | 43.17 | 94.64 | 83.55 | 64.88 | 98.67 | 68.90 | 79.88 | 99.65 | 86.57 | 54.83 | 52.99 | 75.25 ± 0.12 | - |
| LoRA Finetuning | 39.92 | 93.71 | 81.04 | 63.37 | 98.59 | 74.38 | 76.25 | 99.58 | 87.38 | 53.80 | 52.97 | 74.64 ± 0.08 | - |
| CHEEM ($MLP^{Down}$, HEE) | 52.51 | 80.59 | 79.67 | 57.43 | 97.86 | 73.94 | 77.89 | 99.60 | 87.37 | 61.73 | 51.02 | 74.51 ± 0.28 | 1.86 ± 0.04 |
| CHEEM ($MLP^{Down}$, PE) | 53.03 | 80.50 | 80.16 | 57.66 | 97.86 | 80.37 | 78.11 | 99.62 | 85.95 | 62.43 | 49.81 | 75.05 ± 0.12 | 1.85 ± 0.06 |
| CHEEM (Attn Proj, HEE) | 50.65 | 80.35 | 78.44 | 56.77 | 97.75 | 77.23 | 77.71 | 99.55 | 86.84 | 61.72 | 51.27 | 74.39 ± 0.13 | 1.95 ± 0.03 |
| EWC | 37.38 | 13.94 | 48.87 | 0.00 | 83.14 | 0.00 | 50.65 | 93.72 | 30.44 | 2.89 | 27.57 | 35.33 ± 0.32 | 7.34 ± 0.55 |
| CODA-Prompt | 0.00 | 1.77 | 2.75 | 0.04 | 0.32 | 0.00 | 22.46 | 3.94 | 5.80 | 0.27 | 24.45 | 5.62 ± 0.25 | 42.58 ± 0.81 |
| DualPrompt | 0.66 | 42.28 | 59.13 | 3.03 | 42.04 | 0.86 | 42.10 | 55.06 | 47.42 | 5.75 | 41.47 | 30.89 ± 0.29 | 17.53 ± 0.27 |
| L2P | 0.11 | 39.46 | 47.87 | 4.11 | 29.80 | 1.02 | 37.07 | 0.83 | 50.15 | 1.29 | 43.97 | 23.24 ± 0.14 | 25.81 ± 0.37 |
| S-Prompts | 36.00 | 79.08 | 71.58 | 50.50 | 93.87 | 72.27 | 67.97 | 98.66 | 87.44 | 40.01 | 43.22 | 67.33 ± 0.38 | 1.80 ± 0.02 |
| DIKI | 33.95 | 76.57 | 71.13 | 54.84 | 92.66 | 71.79 | 70.46 | 97.40 | 87.61 | 40.13 | 47.41 | 67.63 ± 0.06 | 1.76 ± 0.01 |
| LoRA ($MLP^{Down}$) | 39.48 | 78.89 | 78.11 | 54.38 | 97.80 | 73.66 | 74.80 | 99.58 | 85.88 | 53.53 | 45.61 | 71.06 ± 0.02 | 1.87 ± 0.00 |

Table 10: **VDD**: Full results on VDD benchmark, extending Table 3 in the main text.

| Method | CIFAR | DPed | OGlt | SVHN | UCF | GTSR | Flwr | Airc | DTD | Avg. Acc | Avg. Frgt. |
|---|---|---|---|---|---|---|---|---|---|---|---|
| **ViT Base** | | | | | | | | | | | |
| Full Finetuning | 90.65 | 99.97 | 86.06 | 97.75 | 79.54 | 99.35 | 98.03 | 70.29 | 76.99 | 88.74 ± 0.11 | - |
| LoRA Finetuning | 91.44 | 99.50 | 79.43 | 97.42 | 73.36 | 98.95 | 98.96 | 64.03 | 77.64 | 86.75 ± 0.11 | - |
| CHEEM ($MLP^{Down}$, HEE) | 90.06 | 99.59 | 83.32 | 95.87 | 73.96 | 97.09 | 97.48 | 67.13 | 75.85 | 86.71 ± 0.23 | 0.35 ± 0.02 |
| CHEEM (Attn Proj, HEE) | 88.90 | 99.58 | 83.08 | 96.26 | 74.49 | 97.27 | 97.56 | 70.55 | 76.42 | 87.23 ± 0.22 | 0.34 ± 0.01 |
| EWC | 83.69 | 97.69 | 6.91 | 77.43 | 25.92 | 78.20 | 0.06 | 5.98 | 19.91 | 43.98 ± 1.34 | 5.09 ± 1.14 |
| CODA-Prompt | 37.69 | 1.29 | 6.87 | 54.52 | 2.32 | 49.22 | 39.18 | 7.48 | 25.16 | 24.86 ± 2.19 | 26.11 ± 0.75 |
| DualPrompt | 82.34 | 4.04 | 14.37 | 15.02 | 13.41 | 64.42 | 27.20 | 15.29 | 16.37 | 28.05 ± 0.85 | 3.18 ± 0.51 |
| L2P | 86.64 | 4.98 | 14.75 | 6.63 | 14.19 | 27.89 | 25.59 | 16.71 | 18.12 | 23.94 ± 0.72 | 8.98 ± 0.64 |
| S-Prompts | 88.34 | 99.47 | 57.38 | 94.23 | 55.07 | 87.90 | 98.48 | 53.52 | 72.59 | 78.55 ± 0.09 | 0.36 ± 0.04 |
| DIKI | 86.54 | 98.20 | 57.70 | 63.44 | 52.10 | 72.66 | 36.45 | 53.53 | 72.82 | 65.94 ± 0.05 | 0.11 ± 0.01 |
| LoRA ($MLP^{Down}$) | 90.18 | 99.21 | 79.43 | 96.35 | 73.10 | 97.39 | 98.54 | 64.01 | 76.19 | 86.04 ± 0.11 | 0.34 ± 0.03 |
| **DiET Tiny** | | | | | | | | | | | |
| Full Finetuning | 83.50 | 99.97 | 69.71 | 97.24 | 57.97 | 98.95 | 69.04 | 44.46 | 65.02 | 76.21 ± 0.07 | - |
| LoRA Finetuning | 81.29 | 99.96 | 76.93 | 96.37 | 54.83 | 98.16 | 74.37 | 40.66 | 63.67 | 76.25 ± 0.30 | - |
| CHEEM ($MLP^{Down}$, HEE) | 75.75 | 97.73 | 81.64 | 95.30 | 57.26 | 93.11 | 74.76 | 45.91 | 64.13 | 76.18 ± 0.10 | 1.03 ± 0.01 |
| CHEEM (Attn Proj, HEE) | 74.70 | 97.85 | 80.43 | 95.22 | 57.46 | 93.68 | 75.75 | 46.55 | 62.11 | 75.97 ± 0.36 | 1.09 ± 0.01 |
| EWC | 79.39 | 93.96 | 0.03 | 60.13 | 4.97 | 64.41 | 0.00 | 0.58 | 0.00 | 33.72 ± 0.15 | 1.52 ± 0.08 |
| CODA-Prompt | 2.07 | 0.00 | 0.02 | 1.55 | 0.02 | 0.56 | 0.36 | 0.30 | 5.16 | 1.12 ± 0.08 | 37.56 ± 0.40 |
| DualPrompt | 47.87 | 4.48 | 28.60 | 11.53 | 2.54 | 75.67 | 0.40 | 0.57 | 2.61 | 19.36 ± 0.55 | 10.54 ± 0.49 |
| L2P | 56.24 | 1.38 | 0.80 | 0.26 | 2.15 | 37.43 | 1.24 | 0.17 | 3.90 | 11.51 ± 0.76 | 20.90 ± 1.72 |
| S-Prompts | 68.58 | 97.24 | 46.05 | 85.87 | 43.44 | 80.13 | 74.78 | 36.72 | 58.90 | 65.75 ± 0.27 | 0.90 ± 0.02 |
| DIKI | 65.54 | 97.44 | 44.89 | 45.55 | 40.78 | 64.49 | 72.37 | 34.41 | 59.38 | 58.32 ± 0.05 | 0.62 ± 0.00 |
| LoRA ($MLP^{Down}$) | 74.26 | 97.69 | 76.87 | 94.96 | 52.68 | 93.09 | 73.75 | 40.52 | 62.22 | 74.01 ± 0.34 | 1.07 ± 0.02 |

# E   EXPERIMENT DETAILS

**Pretrained Models**: We initialize the pretrained ViT-B/16 and DEiT-Tiny/16 models from the checkpoint available in `timm`. Both models use a patch size of 16 and a resolution of $224 \times 224$. The ViT-B/16 checkpoint has been pretrained on ImageNet 21k and finetuned on ImageNet1k. The DEiT-Tiny/16 checkpoint has been trained on ImageNet1k. All out experiments use the same checkpoints. We refer the reader to Dosovitskiy et al. (2021) for the architecture details of ViT-B/16 and Touvron et al. (2021) for the architecture details of DEiT-Tiny/16.

Our experiments are done using PyTorch and leverage `timm` for architecture implementation. In all our experiments, we use the Adam optimizer Kingma & Ba (2015) with no weight decay. For

experiments with CHEEM, we use a learning rate of $0.001$, 50 epochs for the supernet training and 20 epochs for finetuning. During supernet training, we use an exploration probability of $\epsilon = 0.3$, and use $\epsilon = 0.5$ during the target network selection to encourage more exploration. We do not perform any data augmentations, and simply resize the images to $224 \times 224$. We modify the implementation provided at `https://github.com/GT-RIPL/CODA-Prompt` to perform experiments on CODA-Prompt, DualPrompts and L2P, and use our own implementations for the other baseline methods. We use a single Nvidia A100 GPU for all our experiments.

### E.1 DETAILS OF THE MTIL BENCHMARK

The MTIL benchmark Zheng et al. (2023) consists of 11 tasks: FGVC-Aircraft Maji et al. (2013), Caltech101 Li et al. (2022b), CIFAR100 Krizhevsky et al. (2009), Describable Textures Cimpoi et al. (2014), EuroSAT Helber et al. (2018), VGG-Flowers Nilsback & Zisserman (2008), Food101 Bossard et al. (2014), MNIST LeCun et al. (1998), Oxford Pets Parkhi et al. (2012), Stanford Cars Gebru et al. (2017), SUN397 Xiao et al. (2010). We use the official training and testing splits provided in the constituent datasets. We use the official validation splits for the evolutionary search, and create our own splits when official split is not provided by randomly sampling 10% of the training dataset.

Table 11: Number of samples in the training, validation, and test sets used in the the experiments on the MTIL benchmark, along with the number of categories.

| Task | #Train | #Validation | #Test | #Classes |
|---|---|---|---|---|
| FGVC Aircraft | 3334 | 3333 | 3333 | 100 |
| Caltech101 | 5465 | 608 | 2604 | 101 |
| CIFAR100 | 45000 | 5000 | 19850 | 100 |
| Describable Textures | 1880 | 1880 | 1880 | 47 |
| EuroSAT | 17010 | 1890 | 8100 | 10 |
| VGG-Flowers | 1020 | 1020 | 6149 | 102 |
| Food-101 | 68175 | 7575 | 25250 | 101 |
| MNIST | 54000 | 6000 | 10000 | 10 |
| Oxford Pets | 3312 | 368 | 3669 | 37 |
| Stanford Cars | 7329 | 815 | 8041 | 196 |
| SUN397 | 17865 | 1985 | 19850 | 397 |

### E.2 DETAILS OF THE VDD BENCHMARK

The VDD benchmark Rebuffi et al. (2017) consists of 10 tasks: ImageNet-1k (Russakovsky et al., 2015), CIFAR100 (Krizhevsky et al., 2009), SVHN (Netzer et al., 2011), UCF101 Dynamic Images (UCF) (Soomro et al., 2012; Bilen et al., 2016), Omniglot (Lake et al., 2015), German Traffic Signs (GTSR) (Stallkamp et al., 2012), Daimler Pedestrian Classification (DPed) (Munder & Gavrila, 2006), VGG Flowers (Nilsback & Zisserman, 2008), FGVC-Aircraft (Maji et al., 2013), and Describable Textures (DTD) (Cimpoi et al., 2014). All the images in the VDD benchmark have been scaled such that the shorter side is 72 pixels. However, for a more realistic evaluation, we reconstruct the VDD benchmark with the original images and splits. Except for UFC101, Omniglot, and Daimler Pedestrian Classification, we use the official train, validation and test splits (when a validation split is not avaiable, we construct a validation split by randomly sampling 10% of the training data.). Due to a lack of high resolution images for UFC101, Omniglot, and Daimler Pedestrian Classification, we use the splits and the images provided by the VDD benchmark and resize the images to $224 \times 224$.

Table 12: Number of samples in the training, validation, and test sets used in the the experiments on the VDD benchmark, along with the number of categories.

| Task | #Train | #Validation | #Test | #Classes |
|---|---|---|---|---|
| ImageNet12 | 1108951 | 123216 | 49000 | 1000 |
| CIFAR100 | 45000 | 5000 | 19850 | 100 |
| SVHN | 65931 | 7326 | 26032 | 10 |
| UCF | 6827 | 758 | 1952 | 101 |
| Omniglot | 16068 | 1785 | 6492 | 1623 |
| GTSR | 23976 | 2664 | 12630 | 43 |
| DPed | 21168 | 2352 | 5880 | 2 |
| VGG-Flowers | 1020 | 1020 | 6149 | 102 |
| FGVC Aircraft | 3334 | 3333 | 3333 | 100 |
| Describable Textures | 1880 | 1880 | 1880 | 47 |

## F   EFFECT OF EXPLORATION PROBABILITY ($\epsilon_1$, $\epsilon_2$)

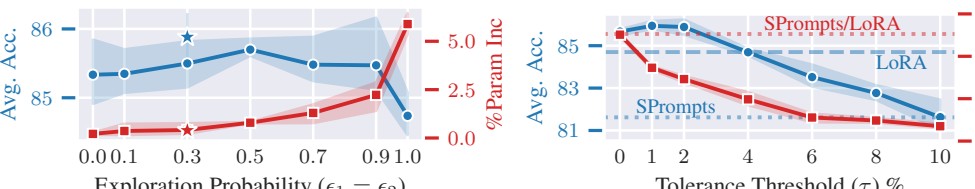

Figure 9a: Effect of the exploration probability on the MTIL benchmark, with exploration probabilities $\epsilon_1$ (supernet training) and $\epsilon_2$ (evolutionary search) set equal. As $\epsilon$ increases, average accuracy first rises, then falls, while the average number of additional parameters per task increases monotonically. This is due to more new operations being learned; $\epsilon = 0.3$ strikes a good balance. **Setting $\epsilon < 0.5$ controls the addition of new operations while maintaining performance.** $\epsilon_1 = 0.3$ and $\epsilon_2 = 0.5$ used in our experiments (denoted by ★) improve accuracy further without increasing parameters. In sum, $\epsilon$ governs number of reuse (exploitation), adapt, and new (exploration) operations.

Figure 9b:   A higher Tolerance Threshold reduces **average FLOPs per task** but also lowers **average accuracy**, as it permits more skip operations to persist in the population during evolutionary search, even if their accuracy is lower (within the tolerance margin). A 2% threshold, used in our experiments, offers a good trade-off. At $\tau = 6\%$, CHEEM still surpasses SPrompts in average accuracy (dotted blue line) while using significantly fewer FLOPs, beyond which the FLOPs plateau. SPrompt FLOPs (dotted red line) closely match those of LoRA, so the same line is used. At $\tau = 4\%$, CHEEM matches LoRA's average accuracy (dashed blue line) with substantially fewer FLOPs. Thus, with $\tau \leq 4\%$, CHEEM matches or exceeds LoRA in accuracy while reducing FLOPs.

## G   THEORETICAL ANALYSIS OF LOCAL VS. GLOBAL ARGMAX OF HEAD CLASSIFIERS IN CONTINUAL LEARNING

### G.1   THE PROBLEM

In continual learning, we have $N$ tasks, each with a different number of classes. Let task $t$ have $C_t$ classes, so by time $T$ we have observed tasks $1, \ldots, T$ with a total of $\sum_{t=1}^{T} C_t$ classes. We train a shared feature extractor $\phi(\mathbf{x}) \in \mathbb{R}^d$ and a growing head classifier composed of task-specific segments $W^t \in \mathbb{R}^{d \times C_t}$.

During training of task $t$, only the segment $W^t$ is updated and used in a softmax over the $C_t$ classes for the current task. However, at inference, for a new test sample $\mathbf{x}$ belonging (in truth) to task $t^*$, the entire head is used: we compute logits for all classes seen so far, and choose the global $\arg\max$. We denote:

- Local argmax:
$$\hat{y}_{\text{local}}(\mathbf{x}) \;=\; \arg\max_{c \in \{1, \ldots, C_{t^*}\}} z_{t^*, c}(\mathbf{x}),$$
  where $z_{t^*, c}(\mathbf{x}) = \langle W^t_{(\cdot, c)}, \phi(\mathbf{x}) \rangle$ are the logits restricted to task $t^*$.

- Global argmax:
$$\hat{y}_{\text{global}}(\mathbf{x}) \;=\; \arg\max_{(t, c) \in \{1, \ldots, T\} \times \{1, \ldots, C_t\}} z_{t, c}(\mathbf{x}).$$

We are interested in the probability that these two predictions coincide:
$$\Pr\big(\hat{y}_{\text{local}}(\mathbf{x}) \;=\; \hat{y}_{\text{global}}(\mathbf{x})\big).$$
Below is a stylized theoretical analysis of why and how often these two can match, highlighting the factors that influence this probability.

### G.2   DISTRIBUTION OF LOGITS AND TASK SEPARATION

Let $z_{t, c}(\mathbf{x})$ be the logit for class $c$ in task $t$ for sample $\mathbf{x}$. We may approximate $z_{t, c}(\mathbf{x})$ by a random variable with mean $\mu_{t, c}$ and variance $\sigma^2_{t, c}$, e.g.,
$$z_{t, c}(\mathbf{x}) \;\approx\; \mu_{t, c} + \epsilon_{t, c}, \quad \epsilon_{t, c} \sim \mathcal{N}(0, \sigma^2_{t, c}).$$

In reality, these means and variances depend on how well the feature $\phi(\mathbf{x})$ and the weights $W^t$ are aligned, but we treat them as parameters to illustrate.

Define:
$$\max_{c \in C_{t^*}} z_{t^*,c}(\mathbf{x}) \quad \text{(the local max for the correct task)}, \tag{7}$$

$$\max_{(t \neq t^*)} \max_{c \in C_t} z_{t,c}(\mathbf{x}) \quad \text{(the max out-of-task logit)}. \tag{8}$$

For $\hat{y}_{\text{local}} = \hat{y}_{\text{global}}$, we need
$$\max_{c \in C_{t^*}} z_{t^*,c}(\mathbf{x}) \geq \max_{(t \neq t^*)} \max_c z_{t,c}(\mathbf{x}).$$

Hence the distribution of all out-of-task logits relative to the best in-task logit is crucial.

## G.3  PROBABILITY OF MATCHING LOCAL AND GLOBAL ARGMAX

### G.3.1  A BASIC TWO-CLASS EXAMPLE

Consider just one class $c^*$ in the true task vs. one class $k$ in an other task. Suppose
$$z_{t^*,c^*} \sim \mathcal{N}(\mu^*, \sigma^2), \quad z_{t',k} \sim \mathcal{N}(\mu', \sigma^2).$$

The probability that $z_{t^*,c^*} \geq z_{t',k}$ is
$$\Pr(z_{t^*,c^*} \geq z_{t',k}) = \Pr(z_{t^*,c^*} - z_{t',k} \geq 0) = \Phi\Big(\frac{\mu^* - \mu'}{\sqrt{2}\,\sigma}\Big),$$

where $\Phi$ is the standard normal CDF.

### G.3.2  MANY CLASSES FROM DIFFERENT TASKS

Now suppose there are $C_{t^*}$ classes in the correct task, and $M = \sum_{t \neq t^*} C_t$ classes outside. Let the local maximum
$$Z^* = \max_{c \in \{1,\ldots,C_{t^*}\}} z_{t^*,c},$$

and let $Z_1, \ldots, Z_M$ represent the logits of the $M$ out-of-task classes. Then
$$\Pr(\hat{y}_{\text{local}} = \hat{y}_{\text{global}}) = \Pr\Big(Z^* \geq \max\{Z_1, \ldots, Z_M\}\Big).$$

If $Z^*$ is (roughly) $\mathcal{N}(\mu_{\text{local}}, \sigma_{\text{local}}^2)$ and each $Z_j$ is $\mathcal{N}(\mu_o, \sigma_o^2)$ (independent simplification), then
$$\Pr\big(Z^* \geq Z_j \text{ for all } j\big) = \int \Big[\Pr(Z_j \leq z)\Big]^M F_{Z^*}(z)\, dz.$$

When $\mu_{\text{local}} > \mu_o$, this probability is high for moderate $M$, but as $M$ grows, the chance that some out-of-task class logit exceeds $Z^*$ increases, unless the gap $\mu_{\text{local}} - \mu_o$ is large.

## G.4  FACTORS INFLUENCING THE MATCH PROBABILITY

1. **Feature Separation Across Tasks.** If $\phi(\mathbf{x})$ strongly separates tasks, then for $\mathbf{x}$ from task $t^*$, out-of-task logits $z_{t,c}$ for $t \neq t^*$ are consistently lower. This increases the probability of $\hat{y}_{\text{local}} = \hat{y}_{\text{global}}$.

2. **Logit Magnitude & Variance.** Even if the means of the correct task's logits exceed those of other tasks, high variance or overlap can cause out-of-task classes to occasionally exceed the correct task's maximum.

3. **Regularization and Task Order.** Continual-learning methods that regularize old task weights or use replay data reduce the chance of weight drift, making it less likely that earlier or other tasks overshadow the correct one.

4. **Task Size Differences.** Larger tasks (more classes) or tasks that were trained earlier might have stronger classifier weights. Conversely, smaller tasks might have very tight, well-separated features. Both can affect how likely a mismatch is.

## G.5  A ROUGH ILLUSTRATIVE BOUND

As a simplistic illustration, suppose:

- For task $t^*$, the local maximum logit $Z^*$ has mean $\mu^*$ and variance $\sigma^{*2}$.

- All out-of-task classes have means $\mu_o < \mu^*$ and variance $\sigma_o^2$.
- There are $M$ out-of-task classes in total.

Then

$$\Pr(\hat{y}_{\text{local}} = \hat{y}_{\text{global}}) \approx \int \Big[\Pr(Z_o \leq z)\Big]^M F_{Z^*}(z)\, dz,$$

where $Z_o$ is the logit distribution for a single out-of-task class and $F_{Z^*}$ is the PDF of $Z^*$. If $\mu^*$ is sufficiently larger than $\mu_o$ (and variances are not too large), $Z^*$ will, with high probability, exceed all $M$ out-of-task logits. But as $M$ grows large, this event can become less likely unless the margin $\mu^* - \mu_o$ is also large.

### G.6 REMARKS

Overall, the probability that the local argmax (over the correct task only) coincides with the global argmax (over all tasks/classes) depends on:

- How well the feature extractor $\phi$ separates tasks, so that out-of-task logits stay low for samples of task $t^*$.
- The relative scale and calibration of classifier weights $W^t$ across tasks.
- The total number of classes from other tasks that could "compete" and produce a large logit by chance.

If tasks are well-separated (and the classifier is carefully regularized or calibrated), this probability can be very high. Conversely, if many classes from older or different tasks produce comparably large logits, the global $\arg\max$ may differ from the local $\arg\max$ more frequently as the number of tasks and classes increases.

Table 13: $\text{Acc}_{Global}$ refers to the average accuracy (Eqn. 4) calculated using the global head, and $\text{Acc}_{Local}$ refers to the same but by masking the logits not belonging to the task. $\text{Acc}_{Train}$ refers to the accuracy calculated after the training on a task is complete, averaged over all the tasks.

| | ViT-B | | | DEiT-Tiny | | |
|---|---|---|---|---|---|---|
| **Method** | $\text{Acc}_{Global}$ | $\text{Acc}_{Local}$ | $\text{Acc}_{Train}$ | $\text{Acc}_{Global}$ | $\text{Acc}_{Local}$ | $\text{Acc}_{Train}$ |
| CODA-Prompt | $40.22 \pm 1.22$ | $79.70 \pm 0.61$ | $86.18 \pm 0.02$ | $5.62 \pm 0.25$ | $34.72 \pm 1.62$ | $67.53 \pm 0.37$ |
| DualPrompt | $33.82 \pm 0.35$ | $83.61 \pm 0.13$ | $84.63 \pm 0.09$ | $30.89 \pm 0.29$ | $68.17 \pm 0.24$ | $71.25 \pm 0.10$ |
| L2P | $26.61 \pm 0.16$ | $80.03 \pm 0.58$ | $84.95 \pm 0.11$ | $23.24 \pm 0.14$ | $60.79 \pm 0.67$ | $71.47 \pm 0.08$ |
| S-Prompts | $81.62 \pm 0.35$ | $84.48 \pm 0.18$ | $84.48 \pm 0.18$ | $67.33 \pm 0.38$ | $70.71 \pm 0.40$ | $70.71 \pm 0.40$ |
| DIKI | $76.42 \pm 0.04$ | $84.50 \pm 0.04$ | $84.50 \pm 0.04$ | $67.63 \pm 0.06$ | $70.86 \pm 0.07$ | $70.86 \pm 0.07$ |
| CHEEM | $85.88 \pm 0.29$ | $88.68 \pm 0.16$ | $88.68 \pm 0.16$ | $74.51 \pm 0.28$ | $78.11 \pm 0.31$ | $78.11 \pm 0.31$ |

## H IDENTIFYING THE TASK-SYNERGY INTERNAL MEMORY IN ViTs

The left of Fig. 1 shows a ViT block. Denote by $x_{L,d}$ an input sequence consisting of $L$ tokens encoded in a $d$-dimensional space. In ViTs, the first token is the so-called class-token, CLS. The remaining $L-1$ tokens are formed by patchifying an input image and then embedding patches, together with additive positional encoding. A ViT block is defined by,

$$z_{L,d} = x_{L,d} + \text{Proj}\Big(\text{MHSA}\big(\text{LN}_1(x_{L,d})\big)\Big), \tag{9}$$

$$y_{L,d} = z_{L,d} + \text{FFN}\Big(\text{LN}_2(z_{L,d})\Big), \tag{10}$$

where $\text{LN}(\cdot)$ represents the layer normalization (Ba et al., 2016), and $\text{Proj}(\cdot)$ is a linear transformation fusing the multi-head outputs from MHSA module. The MHSA realizes the dot-product self-attention between Query and Key, followed by aggregating with Value, where Query/Key/Value are linear transformatons of the input token sequence. The FFN is often implemented by a multi-layer perceptron (MLP) with a feature expansion layer $\text{MLP}^{\text{Up}}$ and a feature reduction layer $\text{MLP}^{\text{Down}}$ with a nonlinear activation function (such as the GELU (Hendrycks & Gimpel, 2016)) in the between, i.e., $\text{FFN}(\cdot) = \text{MLP}^{\text{Down}}\Big(\text{GELU}\big(\text{MLP}^{\text{Up}}(\cdot)\big)\Big).$

The proposed identification process is straightforward. Without introducing any modules handling forgetting, we compare both the task-to-task forward transferrability and the sequential forgetting

Table 14: Ablation studies of identifying where to place our proposed CHEEM in ViT by testing 11 components or composite components (Eqns. 9 and 10).

| Index | Finetuned Component | Avg. Acc. | Avg. Forgetting |
|-------|---------------------|-----------|-----------------|
| 1 | $LN_1 + LN_2$ | 81.76 | 21.24 |
| 2 | FFN | 84.20 | 44.76 |
| 3 | $MLP^{Down}$ | 83.66 | 37.99 |
| 4 | $LN_2$ | 80.04 | 16.35 |
| 5 | $MHSA + LN_1$ | 85.26 | 54.38 |
| 6 | $LN_1$ | 81.18 | 19.04 |
| 7 | Query | 81.57 | 19.69 |
| 8 | Key | 81.56 | 19.19 |
| 9 | Query+Key | 81.49 | 31.10 |
| 10 | Value | 84.99 | 37.58 |
| 11 | Projection | 85.11 | 30.50 |

for different components in a ViT block. **Our intuition is that a desirable component for placing the task-synergy parameter memory must enable strong transferrability with manageable forgetting, while being lightweight to account for the trade-off between stability and plasticity.**

To that end, we use the VDD benchmark (Rebuffi et al., 2017) (see Fig. 5). We first train a ViT-Base (Dosovitskiy et al., 2021) on the first task, ImageNet (Russakovsky et al., 2015), as the base model $F_1(\cdot)$. To measure the task-to-task transferability, we *individually fine-tune $F_1$* in a task-to-task transfer learning manner for the remaining 9 streaming tasks. Let $F_{t|1}$ be the backbone fine-tuned for task $T_t$ (for $t \geq 1$), and $C_t$ the head classifier trained from scratch. The average Top-1 accuracy is defined by Equation 4 where Acc() uses the Top-1 classification accuracy.

To measure the sequential forgetting, we *continually fine-tune* the backbone started from $F_1$ on the 9 tasks in a randomly sampled and fixed streaming order (as shown in Fig. 2a in the main text). Let $F_{1:t}$ be the backbone trained sequentially and continually after task $T_t$ and $H_t$ is its head classifier. The average forgetting (Chaudhry et al., 2018) on the first $N - 1$ streaming tasks is defined by Equation 5, where $a_{j,t} = \text{Acc}(T_t; F_{1:j}, H_t)$.

As shown in Table 14, we compare 11 components or composite components in ViT. Consider the strong forward transfer ability, manageable forgetting, maintaining simplicity and for less invasive implementation in practice, **we select either the Projection layer after the MHSA or the $MLP^{Down}$ as the task-synergy internal (parameter) memory** to realize our proposed CHEEM for ExfCCL (Fig. 1). We test both in experiments and provide ablation studies in Section 3.3.