# OpenReview forum: "Continual Learning by Reuse, New, Adapt and Skip: A Hierarchical Exploration-Exploitation Approach"
_ICLR.cc/2026/Conference — ICLR 2026 Conference Withdrawn Submission_

### Official Review · Reviewer_QXts · 2025-10-27

**Soundness:** 3
**Presentation:** 3
**Contribution:** 3
**Rating:** 6
**Confidence:** 4

**Summary:**

This paper tackles the Exemplar-free Class-incremental Continual Learning (ExfCCL) by identifying the critical "local vs. global argmax" issue as a primary failure mode for existing methods. This paper proposes CHEEM, a novel framework that decouples task-ID inference (via an external memory) from model adaptation (via a dynamic Hierarchical Exploration-Exploitation NAS). The method is innovative and achieves SOTA results on two challenging benchmarks.

**Strengths:**

+ The paper compellingly identifies the "local vs. global argmax" discrepancy as a primary failure mode for existing ExfCCL methods. Even a similar viewpoint has been discussed in previous works [1], the clear diagnosis from theoretical analysis is still interesting.
+ The HEE-NAS mechanism intelligently combines Reuse, New, Adapt (LoRA), and Skip to balance the stability-plasticity dilemma and improves the model efficiency during inference. The similarity-guided HEE sampling strategy (vs. uniform) is well-supported by ablation studies, demonstrating significant parameter savings.
+ The paper demonstrates state-of-the-art performance on the challenging MTIL and VDD benchmarks. The key ablations and the qualitative analysis of learned, efficient architectures (via the Skip operation) effectively support the method's design choices.
- - -
**Reference**
[1] A theoretical study on solving continual learning. In NeurIPS, 2022.

**Weaknesses:**

+ The paper focuses on inference FLOPs (which are low due to the Skip operation) but completely omits any discussion of the training cost. NAS methods, particularly those involving supernet training (Sec 2.3) and evolutionary search (Sec 2.4), are notoriously expensive. The claim of "efficiency" is misleading without this context. The authors must report training complexity (e.g., GPU-hours per task, relative cost vs. baselines) for a fair assessment of the method's practicality.
+ Concerns on the External Memory ($F_1$ Reliance): The entire framework hinges on an external memory built using a single, frozen $F_1$ model. This design appears brittle and its robustness is unverified. It assumes $F_1$ (e.g., trained on ImageNet) can provide a separable feature space for all future tasks, even semantically distant ones. The paper provides no analysis of this component's robustness or its accuracy (e.g., task-ID inference accuracy is not reported), which is a critical point of failure.
+ Section 2.5 states that all New and Adapt modules are randomly re-initialized after the architecture search, discarding the weights learned during the HEE-NAS (Sec 2.3). This appears to contradict the entire purpose of the sophisticated, similarity-guided HEE sampling. If the weights are ultimately discarded, what is the benefit of this complex, guided-sampling-based weight-training process over a simpler search? A critical ablation comparing (A) inheriting supernet weights vs. (B) retraining from scratch is missing.
+ The paper positions itself as a dynamic architecture method. However, it fails to survey or compare against a highly relevant and concurrent line of work: dynamic adapter methods. Techniques like InfLoRA [2] or SDLoRA [3] also dynamically allocate parameters (e.g., LoRA modules) based on task needs, presenting a different and potentially more efficient approach to the same problem. This omission leaves a significant gap in the related work and experimental comparison.
+ The paper's core innovation is the 4-op NAS. However, the ablations are insufficient to isolate each component's contribution. The key comparison (Table 1) is CHEEM (all 4 ops) vs. LORA (1 op). This does not quantify the specific impact of the Skip operation. A CHEEM-without-Skip ablation is necessary to measure the precise trade-off between Skip, accuracy, and inference FLOPs.
- - -
**Reference**
[2] InfLoRA: Interference-free Low-Rank Adaptation for Continual Learning. In CVPR, 2024.
[3] SD-LoRA: Scalable Decoupled Low-Rank Adaptation for Class Incremental Learning. In ICLR, 2025.

**Questions:**

Please refer to the Weaknesses section.

---

### Official Review · Reviewer_wAQH · 2025-10-30

**Soundness:** 3
**Presentation:** 3
**Contribution:** 2
**Rating:** 4
**Confidence:** 4

**Summary:**

This paper proposes a continual learning approach (CHEEM) that uses a hierarchical exploration-exploitation neural architecture search to adapt the architecture using 4 operations (reuse, new, adapt, and skip) and a clustering based task ID inference mechanism. CHEEM is evaluated on MTIL and VDD demonstrating strong performance.

**Strengths:**

1. CHEEM is exemplar-free which helps reduce memory usage and preserve privacy.
2. CHEEM learns adaptive model structures tailored to individual tasks in a semantically meaningful way, which is shown in Fig 2.

**Weaknesses:**

1. A number of recent architectural baselines are missing, e.g., MoE-adapters, prompt-pool.
2. Total number of parameters are not included in experiments (Tables 1 and 3).
3. In most cases, CHEEM only shows modest improvement over LORA, which does not justify the additional steps in CHEEM.

**Questions:**

1. What is the effect of task order on the architecture and performance?
2. How does task relatedness impact the architecture?
3. What is novel about the clustering based a task ID inference mechanism (external task-centroid memory)?

---

### Official Review · Reviewer_v8tT · 2025-10-31

**Soundness:** 3
**Presentation:** 2
**Contribution:** 2
**Rating:** 4
**Confidence:** 3

**Summary:**

This paper addresses the problem of adapting pretrained vision transformers to the exemplar-free class-incremental learning (ExfCCL) setting. Specifically, it proposes CHEEM, which uses neural architecture search (NAS) to determine, for each task, whether to (a) reuse existing layers, (b) adapt existing layers with task-specific LoRA adapters, (c) add a new layer, or (d) skip a layer. Additionally, it keeps track of task centroids to select task-specific classifiers. The authors empirically demonstrate that the proposed method outperforms baselines on two ExfCCL benchmarks, MTIL and VDD, which consist of images from different domains, both in terms of overall accuracy and FLOPs.

Overall, the paper is technically sound, and the proposed method outperforms baselines in both accuracy and the FLOPs of the NAS-searched final model. My main concerns are related to the motivation of the study and the need for additional analysis on training-time complexity and contributions of each component of the method.

**Strengths:**

- The proposed method is intuitive and empirically shown to outperform existing approaches on challenging benchmarks.
- The description of the NAS procedure is clear; however, given the multi-stage nature of the algorithm, pseudocode would be helpful for clarity.

**Weaknesses:**

1. The introduction contains too much detail about the method, making it difficult to identify the core issue in existing methods that this paper addresses. It is unclear under what circumstances one would prefer this relatively complex method over simpler alternatives such as LoRA adapters combined with a task predictor.
2. The task ID inference mechanism---framed as a main contribution---is never explicitly described. Could the authors clarify? Additionally, while the authors note issues with task-agnostic heads, it is unclear why the proposed method performs better than existing approaches that use task predictors or task centroids. Which of the drawbacks listed in Appendix G.4 does the proposed method specifically address?
3. The NAS procedure seems computationally expensive during training, but only the FLOPs of the final model are reported in comparisons.
4. An ablation study on the contribution of each of the four operations would be helpful. For instance, it would be interesting to see the effect of adding "New" on top of "Adapt."

**Questions:**

1. What is the difference between "LoRA finetuning" and "LoRA"?
2. Where is the proposed metric, FoM, used?
3. L52: "no restrictive assumptions": Do the tasks need to have disjoint classes? If not, how does class overlap affect task inference or operation sampling?
4. L468: What exactly are the "substantially significant differences" between the proposed method and L2G, and why is L2G not included in the comparisons?

### Questions/comments that did not impact the score
5. L102: difficult >> difficulty
6.  L104: loca >> local
7. Table 6: VDD-MLP's accuracy is bolded but it is lower than VDD-Attn's.

---

### Official Review · Reviewer_PLw7 · 2025-11-01

**Soundness:** 3
**Presentation:** 2
**Contribution:** 3
**Rating:** 4
**Confidence:** 4

**Summary:**

This paper proposes a new framework for exemplar-free continual learning (ExfCCL) referred to as CHEEM, designed to prevent catastrophic forgetting. It tackles the stability-plasticity dilemma using a Hierarchical Exploration-Exploitation (HEE) approach. A core component, HEE-NAS, acts as an "internal memory" that dynamically adapts the model's architecture by reusing, creating, or adapting components for new tasks. An "external memory" of task centroids is used to infer the correct task ID during testing. CHEEM significantly outperforms state-of-the-art methods on challenging benchmarks by learning adaptive, task-specific model structures.

**Strengths:**

**A Novel Framework**: The paper introduces a new approach by defining Exemplar-Free Continual Learning (ExfCCL) as the problem of managing two separate memories within a Vision Transformer (ViT): an external memory for task identification (task centroids) and an internal memory for model parameters.

**A Dynamic Architecture Method**: It presents a unique Neural Architecture Search (NAS) method that uses hierarchical exploration-exploitation. This method maintains the internal memory by continually learning dynamic, task-aware models. It uses four distinct operations—Reuse, Adapt, New, and Skip—to selectively update the model, which helps mitigate catastrophic forgetting.

**State-of-the-Art Results**: The method achieves top-tier performance on two difficult benchmarks (MTIL and VDD), as measured by a new Figure of Merits (FoM) metric. Crucially, it also demonstrates the ability to automatically learn logical, customized model structures suited for each individual task.

**Weaknesses:**

**Missing Algorithm Details**: The paper's reproducibility and clarity would be significantly improved by including pseudocode that details the step-by-step execution of the proposed CHEEM algorithm within a continual learning scenario.

**Unverified "Skip" Operation**: The contribution of the "Skip" operation is not empirically validated. The experimental results should include an analysis (or ablation study) showing its specific impact on the number of parameters, training time, and inference time.

**Incomplete Results in Main Tables**: To properly demonstrate the parameter efficiency of CHEEM, Tables 1 and 3 should be updated to include the total number of learnable parameters for all compared methods.

**Missing Key Baseline Comparisons**: The evaluation is incomplete. It should include a direct comparison against DualPrompt on the standard 10-Split and 20-Split CIFAR-100 and 10-Split ImageNet-R benchmarks. This comparison must report:

 - The number of learnable parameters.

 - The average accuracy (performance).

 - The forgetting metric (Forget / BWT).

 - The training and inference (test) times.

**Questions:**

Please refer to the weaknesses.

---

### Note · Authors · 2025-11-14

I have read and agree with the venue's withdrawal policy on behalf of myself and my co-authors.